# Association of Four Interleukin-8 Polymorphisms (−251 A>T, +781 C>T, +1633 C>T, +2767 A>T) with Ovarian Cancer Risk: Focus on Menopausal Status and Endometriosis-Related Subtypes

**DOI:** 10.3390/biomedicines12020321

**Published:** 2024-01-30

**Authors:** Rafał Watrowski, Eva Schuster, Gerda Hofstetter, Michael B. Fischer, Sven Mahner, Toon Van Gorp, Stefan Polterauer, Robert Zeillinger, Eva Obermayr

**Affiliations:** 1Department of Obstetrics and Gynecology, Helios Hospital Muellheim, Teaching Hospital of the University of Freiburg, Heliosweg 1, 79379 Muellheim, Germany; rafal.watrowski@gmx.at; 2Faculty of Medicine, University of Freiburg, 79106 Freiburg, Germany; stefan.polterauer@muv.ac.at; 3Molecular Oncology Group, Department of Obstetrics and Gynecology, Comprehensive Cancer Center-Gynaecologic Cancer Unit, Medical University of Vienna, Waehringer Guertel 18–20, 1090 Vienna, Austria; eva.schuster@meduniwien.ac.at (E.S.); robert.zeillinger@muv.ac.at (R.Z.); 4Department of Pathology, Medical University of Vienna, Waehringer Guertel 18–20, 1090 Vienna, Austria; gerda.hofstetter@meduniwien.ac.at; 5Department of Blood Group Serology and Transfusion Medicine, Medical University of Vienna, Waehringer Guertel 18–20, 1090 Vienna, Austria; michael.b.fischer@meduniwien.ac.at; 6Center for Biomedical Technology, Department for Biomedical Research, Danube University Krems, Dr.-Karl-Dorrek-Straße 30, 3500 Krems, Austria; 7Department of Gynaecology, University Medical Center Hamburg-Eppendorf, 20246 Hamburg, Germany; sven.mahner@med.uni-muenchen.de; 8Department of Obstetrics and Gynaecology, University Hospital, Ludwig-Maximilians-University Munich, 81377 Munich, Germany; 9Division of Gynaecologic Oncology, University Hospital Leuven, 3000 Leuven, Belgium; toon.vangorp@uzleuven.be; 10Leuven Cancer Institute, Catholic University of Leuven, 3000 Leuven, Belgium

**Keywords:** interleukin-8, IL-8, single-nucleotide polymorphism, ovarian cancer, endometriosis-related cancer, postmenopause

## Abstract

Interleukin-8 (IL-8) is involved in the regulation of inflammatory processes and carcinogenesis. Single-nucleotide polymorphisms (SNPs) within the IL-8 gene have been shown to alter the risks of lung, gastric, or hepatocellular carcinomas. To date, only one study examined the role of IL-8 SNPs in ovarian cancer (OC), suggesting an association between two IL-8 SNPs and OC risk. In this study, we investigated four common IL-8 SNPs, rs4073 (−251 A>T), rs2227306 (+781 C>T), rs2227543 (+1633 C>T), and rs1126647 (+2767 A>T), using the restriction fragment length polymorphism (PCR-RFLP) technique. Our study included a cohort of 413 women of Central European descent, consisting of 200 OC patients and 213 healthy controls. The most common (73.5%) histological type was high-grade serous OC (HGSOC), whereas 28/200 (14%) patients had endometriosis-related (clear cell or endometrioid) OC subtypes (EROC). In postmenopausal women, three of the four investigated SNPs, rs4073 (−251 A>T), rs2227306 (+781 C>T), and rs2227543 (+1633 C>T), were associated with OC risk. Furthermore, we are the first to report a significant relationship between the T allele or TT genotype of SNP rs1126647 (+2767 A>T) and the EROC subtype (*p* = 0.02 in the co-dominant model). The TT homozygotes were found more than twice as often in EROC compared to other OC subtypes (39% vs. 19%, *p* = 0.015). None of the examined SNPs appeared to influence OC risk in premenopausal women, nor were they associated with the aggressive HGSOC subtype or the stage of disease at the initial diagnosis.

## 1. Introduction

Ovarian cancer (OC) is the deadliest gynecological malignancy, with 5-year survival rates between 40 and 50% [1,2]. In 2019, approximately 300,000 women worldwide were newly diagnosed with OC, and around 200,000 died from the disease [1,2]. Globally, OC accounts for 3.4% of new cancer cases and 4.7% of cancer-related deaths in women [1]. The poor prognosis of OC can be attributed to several factors, including its high biological aggressiveness and heterogeneity, its oligosymptomatic progress until advanced stages, and insufficient prevention strategies [1,2,3,4]. The cornerstone of OC treatment lies in radical cytoreductive surgery, followed by chemotherapy and, eventually, targeted therapies [4,5]. From a histological perspective, OC represents an umbrella term encompassing diverse primary sites, such as the ovary, fallopian tube, and peritoneum, as well as various histotypes [6]. Among these, high-grade serous OC (HGSOC)—developing from the tubal epithelium—stands as the most prevalent histological subtype, accounting for approximately 75% of all OC cases [6]. Endometrioid and clear cell OC—usually considered together as endometriosis-related OC (EROC) subtypes [7,8]—originate from endometriotic lesions and display better clinical outcomes compared to HGSOC [8]. Patients with endometriosis face a 2.5% lifetime risk of developing OC, compared to the general population’s 1.1–1.5% lifetime risk [1,7,8]. EROC patients are more frequently diagnosed at earlier stages and demonstrate improved overall and progression-free survival as compared to non-EROC counterparts [9].

Hereditary factors play a significant role in OC pathogenesis, with BRCA1 and BRCA2 mutations being the most prominent examples [10]. However, at least 15% of hereditary ovarian cancers are derived from a genetic condition unrelated to BRCA genes [11,12]. Given the substantial hereditary component associated with OC, the early identification of individuals at risk may improve the early detection, personalized prevention, and treatment of OC [13]. Single-nucleotide polymorphisms (SNPs) represent the most common type of genetic variation in the human, in which a single nucleotide (A, T, C, or G) is replaced by another nucleotide at a specific position within coding and non-coding sequences. Around 100 SNPs have been identified as genetic factors that may modify an individual’s risk of OC [13]. Some of these variants may specifically alter the risk for certain cancer subtypes [14,15] or have varying relevance depending on life stage, such as pre- or postmenopause [15,16].

Interleukin-8 (IL-8), also known as CXCL8, is a pro-inflammatory and pro-angiogenic chemokine that belongs to the CXC chemokine supergene family [17]. There are two main forms of IL-8, the 72-amino acid monocyte-derived form and the endothelial form, which has 5 extra N-terminal amino acids [17,18]. Various cell types, including neutrophils, monocytes, fibroblasts, endothelial cells, airway smooth muscle cells, and epithelial cells, release IL-8. The biological effects of IL-8 are mediated through the binding of IL-8 to two cell-surface G protein-coupled receptors, referred to as CXCR1 and CXCR2 [17]. Elevated expressions of IL-8 and its receptors are commonly observed in various cell types within the tumor microenvironment, including cancer cells, endothelial cells, infiltrating neutrophils, and tumor-associated macrophages. IL-8 signaling promotes angiogenic responses in endothelial cells, enhanced proliferation, and the survival of endothelial and cancer cells, and it potentiates the migration of cancer cells, endothelial cells, and infiltrating neutrophils at the tumor site [17].

Regarding OC, IL-8 expression is upregulated in cancerous compared to normal ovarian tissues. OC cells and tumor-associated stromal cells produce IL-8 and express CXCR1 and CXCR2 receptors [19,20]. Interferon-gamma-induced expression of the Bcl3 proto-oncogene, with subsequent expression of IL-8 in OC cells, facilitates OC cell migration [20,21]. IL-8 released by cancer-associated fibroblasts has been shown to attenuate autophagy and enhance the migration of OC cells [22]. Furthermore, IL-8 plays a role in the recruitment of tumor-associated neutrophils (TANs) and induces JAG2 expression within TANs [23]. IL-8 enhances the proliferation and survival of OC cells via activation of multiple signaling pathways, including MAPK/ERK, PI3K/Akt, and JAK/STAT mediated signaling cascades [17,24,25,26]. Finally, IL-8 has been implicated in promoting chemoresistance by inhibiting apoptosis and supporting the survival of OC cells [26].

IL-8 is the best-studied chemokine related to endometriosis, a chronic inflammatory condition affecting around 10% of women [27,28]. Serum and peritoneal IL-8 levels correlate with disease severity [28] and endometriosis-associated pain [29]. Inhibition of IL-8 can improve inflammation and fibrosis in endometriosis [30]. A recent study found an association between IL-8 −251T/A (rs4073) polymorphism and chronic pelvic pain in women with endometriosis [31].

The IL-8 gene is located on chromosome 4q12-q135 and encodes a 99-amino acid precursor protein, processed into active IL-8 isoforms. The IL-8 gene consists of four exons and three introns [17,32]. The 5′-flanking region of the IL-8 gene contains several nuclear factor binding sites. Transcription of the IL-8 gene is primarily controlled by nuclear factor-κB (NF-κB) through tumor necrosis factor (TNF) and TNF receptor-associated factor 6 (TRAF6) [17,32]. The functional consequences of polymorphic IL-8 genetic variants arise from two main scenarios: altered gene expression and/or structural modification [32]. SNPs in the IL-8 gene’s promoter region modify its expression. The altered expression level of the IL-8 gene regulates the magnitude of the pro-inflammatory response and is associated with various disease phenotypes. The second scenario involves structural changes in the receptor binding sites of the IL-8 protein, affecting its binding to its receptors (CXCR1 and CXCR2) and, thus, impacting IL-8-mediated cell signaling and the activation of inflammatory cells [17,32].

In total, 734 SNPs were mapped in the human IL-8 gene sequence, including 21 in the promoter region, 41 in the coding region, 137 in the intron regions, and 100 SNPs in the 3′ UTR [32]. By far, the most studied Il-8 SNP is the −251 A>T (rs4073) in the promoter region (rs4073), followed by the +781 C/T (rs2227306) in intron 1 [32,33,34,35]. The IL-8 −251A>T (rs4073) polymorphism, located in the promoter region, has been associated with increased IL-8 levels [14,36,37]. The −251T allele had two to five times stronger transcriptional activity than the −251A allele [14]. In patients with acute coronary syndrome, IL-8 plasma levels correlated with the IL-8 251 A>T polymorphism; the highest IL-8 plasma concentrations were found in carriers of the IL-8 −251 AA genotype, intermediate levels among carriers of the −251 AT genotype, and the lowest levels in individuals carrying the −251 TT genotype [38]. Additionally, the IL-8 +781C>T (rs2227306) polymorphism, located in the first intron, has been described to enhance gene transcription and regulation [37].

Despite an extensive body of literature on the −251 A>T SNP, the precise functional implications of other IL-8 SNPs remain elusive. Generally, SNPs within the promoter region (e.g., rs4073 −251A>T) impact gene expression by modifying promoter activity, influencing the binding of transcription factors, altering DNA methylation patterns, and affecting histone modifications [39]. SNPs located within intronic regions (e.g., rs2227306, +781 C>T or rs2227543, +1633 C>T) induce splice variants of transcripts, thereby either promoting or disrupting the binding and function of long noncoding RNAs. Lastly, SNPs situated in the 3′-UTR (e.g., rs1126647, +2767 A>T) exert their influence on microRNA (miRNA) binding [39]. Hacking et al. [36] suggested that the influence on IL-8 production is likely attributed to specific haplotypes comprising multiple SNPs, including −251A>T, +396G>T, +781C>T, +1238delA/insA, +1633C>T, and +2767A>T. These SNPs collectively form haplotype 2 (A/G/T/delA/T/T), which is associated with significantly higher IL-8 transcription levels compared to haplotype 1 [36].

Appropriately, given their roles in the regulation of the immunologic response, IL-8 SNPs (particularly the −251A>T; rs4073) have been linked with increased susceptibility to asthma [37], acute pancreatitis [40], rheumatoid arthritis [41], acute coronary syndrome [38], and macular degeneration [42]. Increased cancer risk was associated with the −251A>T SNP overall, under the models of A allele vs. T allele, AA vs. TT, and AA vs. AT/TT [34]. With regard to selected malignancies, SNPs within the IL-8 gene have been shown to decrease the risks of hepatocellular [35,43] and nasopharyngeal carcinomas [44], and to increase the susceptibility to gliomas [45] and osteosarcomas [46], as well as gastric [47], lung [48,49], prostate [50], and some subsets of breast carcinomas [51,52,53].

To date, only one study examined the role of IL-8 SNPs in ovarian cancer (OC), suggesting an association between two IL-8 SNPs (IL-8 +781 and IL-8 +2767) and OC risk [54]. Therefore, the association between IL-8 SNPs and ovarian cancer (OC) risk remains understudied.

In the present study, we investigated the relationships between four common IL-8 polymorphisms and OC risk. We additionally analyzed the role of the IL-8 SNPs in relating menopausal status to the most relevant OC subtypes, HGSOC, and EROC.

## 2. Materials and Methods

This study utilized a case–control design. We examined four common SNPs within the IL-8 gene, rs4073 (−251 A>T), rs2227306 (+781 C>T), rs1126647 (+2767 A>T), and rs2227543 (+1633 C>T), using the restriction fragment length polymorphism (PCR-RFLP) technique in blood samples from 413 women of Central European descent, of whom 200 women were diagnosed with OC and 213 were healthy female controls.

All blood samples were retrieved from a blood bank at the Molecular Oncology Lab of the Medical University of Vienna. All samples from this collection were obtained from patients and controls recruited from 1996 to 2021 at the Medical University of Vienna and collaborating European institutions. For the present study, we selected samples exclusively from women of Central European descent, including those from Austria, Poland, Germany, and Belgium. All samples from the blood bank were obtained from patients and controls who gave their written consent. The blood bank project (EK-366/2003, EK1966/2020) and the analysis of SNPs in OC risk (EK-293/2011) were both approved by the Ethics Committee of the Medical University of Vienna. Patient-specific and clinical–pathological data were stored anonymously in a database and handled according to the principles of good scientific practice. Clinicopathological classification and staging were performed in accordance with the WHO (2014) [55] and FIGO (2013) [56] classifications. The age of 51 was used as a proxy for menopausal status, corresponding to the mean age of menopause in Austria [57] and Central Europe [58].

### 2.1. DNA Extraction and Genotyping

Peripheral blood was collected from all participants in EDTA tubes. Genomic DNA was isolated from white blood cells using the QIAamp DNA Blood Mini Kit (QIAGEN). IL-8 polymorphisms were determined by analyzing fragment length polymorphisms of the respective PCR products (PCR-RFLP). The amplicons were generated from 25 ng genomic DNA as template in a 25 µL reaction mix, containing 5 pmol of the respective forward and reverse primers (Table 1) and MangoMix™ (Bioline) providing MangoTaq™ DNA polymerase, MgCl_2_, and dNTPs. The amplification was carried out after an initial hot start at 95 °C for 5 min, for 45 cycles starting with a 30 s denaturation at 95 °C, followed by a 30 s annealing at the temperature given in Table 1, and a 60 s extension at 72 °C. After a final extension step at 72 °C for 7 min, the PCR products were digested with the respective restriction endonuclease (all from New England Biolabs, Ipswich, MA, USA) under the conditions given in Table 1. The restriction fragments were separated with capillary electrophoresis using the Fragment Analyzer™ Automated CE System (Advanced Analytical, Ankeny, IA, USA) and the DNF-905 dsDNA Kit (Agilent, Santa Clara, CA, USA). The sizes of the fragments were assessed using the software PROSize^®^ 3.0 version 3.0.1.6 (Advanced Analytical Technologies).

### 2.2. Statistical Analysis

All statistical analyses were carried out with JASP statistical software v.0.17.3 for Windows [59] and the VassarStats Website for Statistical Computation [60]. The χ^2^ test, with one or two degrees of freedom, and the Fisher’s exact test were used to examine differences in genotype and allele frequencies between patients and controls. Odds ratios (ORs), with 95% confidence intervals (CIs), were calculated to assess the effect of each SNP on OC risk. The Hardy–Weinberg equilibrium (HWE) of the four polymorphisms in the control group was tested using a goodness-of-fit χ^2^ test. Differences in age between the study groups were assessed using the Student’s t-test. A two-tailed *p*-value ≤ 0.05 was considered statistically significant.

We explored the association of OC risk or traits using following models:Co-dominant (general test of association): DD versus Dd versus dd;Dominant: (DD + Dd) versus dd;Recessive: DD versus (Dd + dd);Overdominant (heterozygote superiority) model: Dd versus (DD + dd);Heterozygote comparison: Dd vs. DD;Homozygote comparison: DD vs. dd;Allelic/multiplicative (allelic frequency): D versus d;

where D is the minor allele and d is the major allele [61,62].

## 3. Results

The mean age of patients in the case and control groups were 55.8 (SD 12.3) years and 51.4 (SD 13.3) years, respectively. The proportion of postmenopausal patients (defined as age ≥ 51 years) did not significantly differ (*p* = 0.12) between the cases (62%; n = 124) and controls (54.5%; n = 116).

Most OC patients (84.5%; n = 169) were diagnosed with advanced FIGO stages (IIb-IV). The most common histological type was HGSOC (73.5%), with 28 out of 200 patients (14%) presenting with EROC subtypes (clear cell or endometrioid). Further details regarding the study population are summarized in Table 2, with the distribution of histological subtypes provided in Table A1, the detailed FIGO stages in Table A2, and the age distribution visualized in Figure A1 (Appendix A).

### 3.1. Single-Nucleotide Polymorphisms (SNPs) and OC Risk

The images of the fragment analyzer electropherograms are provided in Appendix A (Figure A2, Figure A3, Figure A4 and Figure A5). As shown in Table 3, none of the four SNPs influenced the overall risk of OC in the unstratified cohort, although carriers of the A allele of SNP rs4073 (−251 A>T) (52.3% vs. 46.7%, *p* = 0.11), and those with the T allele of rs2227306 (+781 C>T) (47.7% vs. 43.7%, *p* = 0.09) were more common among cases. None of the examined SNPs were associated with the aggressive HGSOC subtype or the stage of disease (early vs. advanced) at the initial diagnosis.

#### 3.1.1. Postmenopausal Women and SNPs

After stratification of the studied cohort according to menopausal status, numerous significant results could be observed in postmenopausal women (see Table 4).

In postmenopausal women, the A allele of rs4073 (−251 A>T) was significantly more common in patients diagnosed with OC (52.8% vs. 43.1%, *p* = 0.03). The association between rs4073 and OC was most striking in the dominant model (*p* = 0.02), confirming that the presence of at least one A allele is linked to a higher risk of OC. Finally, when comparing AA homozygotes to TT homozygotes, the OR for OC in those with the AA genotype increased to 2.07, suggesting a more than double risk of OC associated with this genotype, but with a relatively wide confidence interval (CI 95% 1.02–4.2) and a borderline *p*-value of 0.05.

Similarly, the T allele of rs2227306 (+781 C>T) was found to be significantly more common in postmenopausal women diagnosed with OC, compared to the control group (48.8% vs. 39.2%, *p* = 0.03). Furthermore, this association remained significant when analyzed in the dominant model (*p* = 0.044), indicating that the presence of at least one T allele is linked to a higher risk of OC.

Significant results were obtained for rs2227543 (+1633 C>T), where the T allele was more than 10% prevalent in postmenopausal OC cases as compared to controls (48.8% vs. 38.4%, *p* = 0.02), with a relevant association of the genotypes and OC risk noted in the dominant model (*p* = 0.016). Being CC homozygous appeared to reduce the risk of OC, as the prevalence of CC homozygotes was significantly higher in controls (42.2% vs. 26.6%, *p* = 0.019). Notably, none of these associations could be observed in women before menopause.

#### 3.1.2. Association between rs2227543 (+1633 C>T) and EROC

A novel and important finding of our study was the significant association between the rs2227543 (+1633 C>T) SNP and the EROC subtype (*p* = 0.044 for the co-dominant model). As shown in Table 5, the TT homozygotes were found more than twice in EROC compared to other OC subtypes (39% vs. 19%, *p* = 0.015), and the CC homozygotes were more prevalent in healthy controls (41.4% vs. 26.6%, *p* = 0.049). The strong association was most visible in the dominant model (χ^2^ 5.85, *p* = 0.016). This association is reported here for the first time.

## 4. Discussion

OC is known for its intense interaction with the immune system. It is capable of triggering a systemic acute inflammatory reaction [63,64]. Interleukins are the key mediators in systemic changes associated with OC [65]. Nevertheless, the evaluation of SNPs in specific interleukins, and their associations with OC risk, remains relatively limited, focusing on only a few ILs, such as IL-1, IL-6, IL-23 or IL-31, and yielding contradictory results [66,67,68,69,70]. To date, only one study has examined IL-8 SNPs and their relationship with OC risk, reporting associations between the IL-8 +781 (T/T) genotype (*p* = 0.005) and increased susceptibility to OC, compared to CC homozygotes. Additionally, an association of the IL-8 +2767 TT genotype with a higher risk of ovarian cancer (*p* = 0.018) was found in a combined German/Moldavian cohort. Interestingly, no association was observed for the most commonly studied IL-8 SNP, rs 4073 (−251A>T) [54].

In the present research, we identified that three out of four investigated SNPs, rs4073 (−251 A>T), rs2227306 (+781 C>T), and rs2227543 (+1633 C>T), were associated with OC risk in postmenopausal patients, but not in the premenopausal group. The high proportion of premenopausal women among both cases and controls, 38% (76/200) and 45.5% (97/213), explains the diminishing significance of observed associations in relation to the entire cohort.

Menopause represents a significant shift in endocrine and immunological homeostasis, potentially altering the impact of genetic factors. Notably, IL-8 plays a pivotal role in the aging process [71,72]. In a study by Shin et al., which aimed to identify a proteomic signature of chronological age and menopause, a close relationship between menopausal age, years since menopause, and plasma IL-8 levels was found [71]. Hot flashes, a phenomenon typically heralding menopause, are strongly associated with circulating IL-8 and TNF-α levels, highlighting the link between increasing systemic inflammation and aging/menopause [72]. The observation that certain SNPs exert their impacts either before or after menopause is not unique to our study. For example, Wang et al. reported significant associations of the TT genotype in the IL-8 (−352A>T) polymorphism with breast cancer risk, but this effect was only significant in postmenopausal women [53]. In another study, AT and TT genotypes at IL-8 rs4073 were associated with reduced breast cancer risk in women aged <55 years [15]. This shift in the physiological role of IL-8 based on menopausal status is not exclusive to malignancy. Sturgeon et al. documented significant relationships between IL-8 levels, depression levels, and pain perception (including pain catastrophizing and pain anxiety), exclusively in postmenopausal women, and not in their premenopausal counterparts [73]. One limitation of our study is that we could not determine whether the SNPs had similar effects, such as on IL-8 expression, in both pre- and postmenopausal women, or if their biological effects differed between these two stages. We hypothesize that the functional consequences of changes in the IL-8 gene structure (such as altered expression or receptor binding) may be overridden by compensatory mechanisms, which might be more effective before menopause. Alternatively, the cumulative impacts of aging and altered immunoregulation may only produce clinical effects after menopause. In summary, our results underscore the importance of considering menopausal status in genetic association studies and emphasize the need for further research involving larger patient populations.

Endometriosis—a chronic inflammatory condition affecting at least 10% of women—exhibits overlapping features with malignant tumors, including local and distant invasion, angiogenesis induction, apoptotic resistance, and stimulation of the inflammatory system [8]. IL-8 is upregulated in endometriotic tissues, promoting the growth and inhibiting the apoptosis of ectopic endometrial cells [30,74]. Despite intense research on IL-8 in endometriosis, the role of genetic variants of IL-8 in the development of EROC has not been explored before.

In this study, we identified a unique association between the presence of the T allele or TT genotype of the rs2227543 (+1633 C>T) and EROC subtypes. TT homozygotes were significantly more prevalent in EROC compared to in other OC subtypes (39.3% vs. 18.5%). While this observation is intriguing, it serves as an initial stepping stone for further investigations. Due to the retrospective nature of our study, we were unable to access information about the history of endometriosis in the study cohort. Furthermore, clear cell OC and endometrioid OC are commonly grouped as EROCs because endometriosis history is noted in 21–51% of CC and 23–43% of endometrioid OC cases [8]. However, endometriosis is not an obligatory precursor for EROC subtypes, and carcinogenesis likely occurs through distinct pathways in each subtype. Clear cell histotypes may arise from pre-existing endometriosis, resulting from retrograde menstruation, while endometrioid OC may originate from ovarian Mullerian metaplasia [8]. Moreover, the risk associated with endometriosis varies depending on non-genetic factors such as premenopausal status (≥45 years) at the time of endometriosis diagnosis, nulliparity, hyperestrogenism, solid compartments, as well as larger size (≥9 cm) of ovarian endometriomas [75]. Considering that only a subset of EROC subtypes originates from endometriosis through multiple pathways, it is reasonable to infer that the observed association with rs1126647 (+2767 A>T) is likely more linked to the initiation of carcinogenesis rather than solely the preexistence of endometriosis.

Although our study provides valuable insights, it is not without limitations. Firstly, the moderate sample size (413 participants) may have affected its statistical power, particularly in subgroup analyses. Secondly, our study was conducted in a specific population with Central European ancestry. While this homogeneity allowed for better control of potential confounding factors, it may also limit the generalizability of our findings to other populations with different genetic backgrounds. Thirdly, we did not assess the relationships between the examined IL-8 SNPs and the circulating levels or biological action of IL-8 in the studied cohort, which limits the insights into the mechanism of the observed associations. Finally, like many genetic association studies, our research inherently relies on retrospective data and, as such, cannot establish causation, but rather, associations. Future studies with larger cohorts are warranted to confirm and extend our findings.

## 5. Conclusions

Our study sheds light on the previously understudied relationship between IL-8 gene SNPs and OC. The main strengths of our study lie in the comprehensive analysis of four different SNPs, an ethnically well-defined cohort, a stratified approach with consideration of menopausal status, and the discovery of a unique association between the EROC and rs1126647. Therefore, our findings can contribute to improved risk assessment and early detection of OC, particularly in postmenopausal women, and highlight a potential genetic marker for EROC. However, further research is warranted to validate these associations in larger and more diverse cohorts and to elucidate the underlying mechanisms driving these relationships, which could have implications for OC risk assessment and personalized prevention strategies.

## Figures and Tables

**Table 1 biomedicines-12-00321-t001:** PCR-RFLP of IL-8 SNPs. Primers for amplification, annealing temperature, and restriction enzyme for digestion.

SNP	Symbol	Location	Primer Sequence	Annealing Temperature	Digestion (Enzyme, Temperature, Duration)	Fragment Size (bp)
−251 (T/A)	rs4073	Promoter	Forward: 5′-TCATCCATGATCTTGTTCTAA-3′ Reverse: 5′-GGAAAACGCTGTAGGTCAGA-3′	55 °C	Mfe I, 37 °C, 25 min	T/T: 524 A/A: 449,75
+781 (C/T)	rs2227306	Intron 1	Forward: 5′-CTCTAACTCTTTATATAAGGAATT-3′ Reverse: 5′-GATTGATTTTATCAACAGGCA-3′	50 °C	EcoR I, 37 °C, 25 min	T/T: 203 C/C: 184,19
+1633 (C/T)	rs2227543	Intron 3	Forward: 5′-CTGATGGAAGAGAGCTCTGT-3′ Reverse: 5′-TGTTAGAAATGCTCTATATTCTC-3′	55 °C	NIa III, 55 °C, 35 min	T/T: 397 C/C: 234,163
+2767 (A/T)	rs1126647	3′UTR	Forward: 5′-CCAGTTAAATTTTCATTTCAGGTA-3′ Reverse: 5′-CAACCAGCAAGAAATTACTAA-3′	50 °C	BstZ17I, 37 °C, 25 min	A/A: 222 T/T: 198,24

3′UTR—3′ untranslated region.

**Table 2 biomedicines-12-00321-t002:** Study population characteristics.

Parameter		Cases	Controls	*p*
Number of individuals		200	213	

Mean age at diagnosis (years)		55.85 (SD 12.3, IQR 47–65)	51.44 (SD 13.3, IQR 45–58)	<0.001

Menopausal status	Postmenopausal	124 (62%)	116 (54.5%)	0.12
(age < 51 vs. ≥51 years)	Premenopausal	76 (38%)	97 (45.5%)	

High-grade serous OC	HGSOC	147 (73.5%)		
	Non-HGSOC	46 (23%)		
	N/a	7 (3.5%)		

EROC	EROC	28 (14%)		
	Non-EROC	165 (82.5%)		
	N/a	7 (3.5%)		

Stage	Early	23 (11.5%)		
	Advanced	169 (84.5%)		
	N/a	8 (4%)		


N/a—non-available for analysis (e.g., only grading available or stage missing), SD—standard deviation, IQR—interquartile range (25–75%), EROC—endometriosis-related OC.

**Table 3 biomedicines-12-00321-t003:** Genotype and allele frequencies of IL-8 gene polymorphisms among OC cases and healthy controls.

SNP	Model	Genotype	Controls	Cases	OR (95% CI)	P ^Fi^	χ^2^	P ^Chi^
**rs4073**	Co-dominant	AA	45 (21.1%)	53 (26.5%)			2.6	0.272
**(−251 A>T)**		AT	109 (51.2%)	103 (51.5%)				
		TT	59 (27.7%)	44 (22%)				
pHWE = 0.78								
	Dominant	AA + AT	154 (72.3%)	156 (78%)	1.36 (0.87–2.13)	0.21	1.79	0.181
		TT	59 (27.7%)	44 (22%)				
	Recessive	AA	45 (21.1%)	53 (26.5%)	1.35 (0.85–2.12)	0.21	1.65	0.200
		AT + TT	168 (78.9%)	147 (73.5%)				
	Overdominant	AT	109 (51.2%)	103 (51.5%)	1.01 (0.69–1.49)	1	0.004	0.95
		AA + TT	104 (48.8%)	97 (48.5%)				
	Homozygote	AA	45 (43.3%)	53 (54.6%)	1.58 (0.90–2.76)	0.12	2.6	0.107
	(AA vs. TT)	TT	59 (56.7%)	44 (45.4%)				
	Heterozygote	AT	109 (70.8%)	103 (66%)	0.80 (0.5–1.3)	0.394	0.81	0.368
	(AT vs. AA)	AA	45 (29.2%)	53 (34%)				
MAF = 0.47	Allele frequency	A	199 (46.7%)	209 (52.3%)	1.25 (0.95–1.64)	0.125	2.53	0.111
	(A vs. T)	T	227 (53.3%)	191 (47.7%)	0.8 (0.61–1.05)	0.13	2.31	
**rs2227306**	Co-dominant	CC	64 (30%)	53 (26.5%)			1.61	0.447
**(+781 C>T)**		CT	112 (52.6%)	103 (51.5%)				
		TT	37 (17.4%)	44 (22%)				
pHWE = 0.4								
	Dominant	TT + CT	149 (70%)	147 (73.5%)	1.19 (0.78–1.83)	0.45	0.64	0.424
		CC	64 (30%)	53 (26.5%)				
	Recessive	TT	37 (17.4%)	44 (22%)	1.34 (0.82–2.18)	0.265	1.4	0.236
		CT + CC	176 (82.6%)	156 (78%)				
	Overdominant	CT	112 (52.6%)	103 (51.5%)	0.96 (0.65–1.41)	0.844	0.05	0.823
		TT + CC	101 (47.4%)	97 (48.5%)				
	Homozygote	TT	37 (36.6%)	44 (45.4%)	1.44 (0.81–2.54)	0.248	1.56	0.212
		CC	64 (63.4%)	53 (54.6%)				
	Heterozygote	CT	112 (75.2%)	103 (70.1%)	0.773 (0.463–1.291)	0.362	0.97	0.325
	(CT vs. TT)	TT	37 (24.8%)	44 (29.9%)				
	Allele frequency	C	240 (56.3%)	209 (52.3%)	1.32 (0.96–1.82)	0.102	2.85	0.091
MAF = 0.44	T vs. C	T	186 (43.7%)	191 (47.7%)				
**rs2227543**	Co-dominant	CC	70 (32.9%)	56 (28%)			1.39	0.500
**(+1633 C>T)**		CT	104 (48.8%)	101 (50.5%)				
		TT	39 (18.3%)	43 (21.5%)				
pHWE = 1								
	Dominant	TT + CT	144 (67.6%)	144 (72.0%)	1.23 (0.81–1.88)	0.34	0.94	0.331
		CC	69 (32.4%)	56 (28.0%)				
	Recessive	TT	39 (18.3%)	43 (21.5%)	1.22 (0.75–1.98)	0.460	0.66	0.417
		CT + CC	174 (81.7%)	157 (78.5%)				
	Overdominant	CT	104 (48.8%)	101 (50.5%)	1.07 (0.73–1.57)	0.768	0.12	0.729
		TT + CC	109 (51.2%)	99 (49.5%)				
	Homozygote	TT	39 (35.8%)	43 (44.4%)	1.38 (0.79–2.41)	0.32	1.27	0.259
		CC	70 (64.2%)	56 (56.6%)				
	Heterozygote	CT	104 (72.7%)	101 (70.1%)	0.88 (0.53–1.47)	0.695	0.24	0.624
		TT	39 (27.3%)	43 (29.9%)				
	Allele frequency	C	244 (57.3%)	213 (53.3%)	1.18 (0.89–1.55)	0.263	1.35	0.245
MAF = 0.43	T vs. C	T	182 (42.7%)	187 (46.8%)				
**rs1126647**	Co-dominant	AA	73 (34.3%)	56 (28%)			2.51	0.285
**(+2767 A>T)**		AT	105 (49.3%)	102 (51%)				
		TT	35 (16.4%)	42 (21%)				
pHWE = 0.89								
	Dominant	TT + AT	140 (65.7%)	144 (72%)	1.34 (0.88–2.04)	0.20	1.89	0.169
		AA	73 (34.3%)	56 (28%)				
	Recessive	TT	35 (16.4%)	42 (21%)	1.35 (0.82–2.22)	0.256	1.42	0.234
		AT + AA	178 (83.6%)	158 (79%)				
	Overdominant	AT	105 (49.3%)	102 (51%)	1.07 (0.73–1.57)	0.767	0.12	0.729
		TT + AA	108 (50.7%)	98 (49%)				
	Homozygote	TT	35 (32.4%)	42 (42.9%)	1.56 (0.89–2.76)	0.149	2.4	0.121
	TT vs. AA	AA	73 (67.6%)	56 (57.1%)				
	Heterozygote	AT	105 (75%)	102 (70.8%)	0.81 (0.48–1.37)	0.5046	0.62	0.431
		TT	35 (25%)	42 (29.2%)				
	Allele frequency	A	251 (58.9%)	214 (53.5%)	1.25 (0.95–1.64)	0.123	2.46	0.116
MAF= 0.41	T vs. A	T	175 (41.1%)	186 (46.5%)				

pHWE—*p* value for the Hardy–Weinberg Equilibrium, MAF—minor allele frequency, P ^Fi^**—***p* value in Fisher’s exact test, P ^Chi^**—***p* value in Chi-squared test (for df = 1 or df = 2).

**Table 4 biomedicines-12-00321-t004:** Genotype and allele frequencies in postmenopausal women. Significant associations are indicated in bold.

SNP	Model	Genotype	Controls	Cases	OR (95% CI)	P ^Fi^	χ^2^	P ^Chi^
**rs4073**	Co-dominant	AA	25 (21.5%)	34 (27.4%)			5.49	0.06
**(−251 A>T)**		AT	50 (43.1%)	63 (50.8%)				
		TT	41 (35.3%)	27 (21.8%)				
	Dominant	AA + AT	75 (64.7%)	97 (78.2%)	1.96 (1.11–3.48)	0.02	5.44	**0.02**
		TT	41 (35.3%)	27 (21.8%)				
	Recessive	AA	25 (21.5%)	34 (27.4%)	1.38 (0.76–2.49)	0.3	1.11	0.29
		AT + TT	91 (78.5%)	90 (72.6%)				
	Overdominant	AT	50 (43.1%)	63 (50.8%)	1.36 (0.82–2.27)	0.25	1.43	0.23
		AA + TT	66 (56.9%)	61 (49.2%)				
	Homozygote	AA	25 (37.9%)	34 (55.7%)	2.07 (1.02–4.2)	0.051	4.06	**0.04**
		TT	41 (62.1%)	27 (44.3%)				
	Heterozygote	AT	50 (66.7%)	63 (64.9%)	0.93(0.49–1.75)	0.872	0.06	0.806
		AA	25 (33.3%)	34 (35.1%)				
MAF = 0.43	Allele frequency	A	100 (43.1%)	131 (52.8%)	1.48 (1.03–2.12)	0.036	4.54	**0.033**
	(A vs. T)	T	132 (56.9%)	117 (47.2%)				
**rs2227306**	Co-dominant	CC	44 (37.9%)	32 (25.8%)			4.58	0.10
**(+781 C>T)**		CT	53 (45.7%)	63 (50.8%)				
		TT	19 (16.4%)	29 (23.4%)				
	Dominant	TT + CT	72 (62.1%)	92 (74.2%)	1.76 (1.01–3.05)	0.052	4.07	**0.044**
	(DD, Dd) vs. dd	CC	44 (37.9%)	32 (25.8%)				
	Recessive	TT	19 (16.4%)	29 (23.4%)	1.56 (0.82–2.97)	0.198	1.84	0.174
	DD vs. (Dd, dd)	CT + CC	97 (83.6%)	95 (76.6%)				
	Overdominant	CT	53 (45.7%)	63 (50.8%)	1.23 (0.74–2.04)	0.44	0.63	0.427
		TT + CC	63 (54.3%)	61 (49.2%)				
	Homozygote	TT	19 (30.2%)	29 (47.5%)	2.1 (1.01–4.38)	0.065	3.95	**0.047**
		CC	44 (69.8%)	32 (52.5%)				
	Heterozygote	CT	53 (73.6%)	63 (68.5%)	0.78 (0.39- 1.54)	0.4945	0.51	0.475
		TT	19 (26.4%)	29 (31.5%)				
	Allele frequency	C	141 (60.8%)	127 (51.2%)	1.48 (1.03–2.12)	0.0429	4.45	**0.035**
MAF = 0.39	T vs. C	T	91 (39.2%)	121 (48.8%)				
**rs2227543**	Co-dominant	CC	49 (42.2%)	33 (26.6%)			6.51	**0.039**
**(+1633 C>T)**		CT	45 (38.8%)	61 (49.2%)				
		TT	22 (19%)	30 (24.2%)				
	Dominant	TT + CT	68 (58.6%)	91 (73.4%)	1.95 (1.13–3.35)	0.02	5.85	**0.016**
	(DD, Dd) vs. dd	CC	48 (41.4%)	33 (26.6%)				
	Recessive	TT	22 (19%)	30 (24.2%)	1.36 (0.73–2.54)	0.350	0.97	0.326
	DD vs. (Dd, dd)	CT + CC	94 (81%)	94 (75.8%)				
	Overdominant	CT	45 (38.8%)	61 (49.2%)	1.53 (0.91–2.55)	0.119	2.63	0.105
		TT + CC	71 (61.2%)	63 (50.8%)				
	Homozygote	TT	22 (31%)	30 (47.6%)	2.025 (1.00–4.1)	0.053	3.89	**0.049**
		CC	49 (69%)	33 (52.4%)				
	Heterozygote	CT	45 (67.2%)	61 (67%)	0.99 (0.51–1.95)	1	0	1
		TT	22 (32.8%)	30 (33%)				
	Allele frequency	C	143 (61.6%)	127 (51.2%)	0.65 (0.45–0.94)	0.027	5.3	**0.021**
MAF = 0.38	T vs. C	T	89 (38.4%)	121 (48.8%)				
**rs1126647**	Co-dominant	AA	48 (41.4%)	37 (29.8%)			3.59	0.166
**(+2767 A>T)**		AT	50 (43.1%)	62 (50%)				
		TT	18 (15.5%)	25 (20.2%)				
	Dominant	TT + AT	68 (58.6%)	87 (70.2%)	1.66 (0.97–2.83)	0.079	3.49	0.062
	(DD, Dd) vs. dd	AA	48 (41.4%)	37 (29.8%)				
	Recessive	TT	18 (15.5%)	25 (20.2%)	1.37 (0.71–2.68)	0.401	0.88	0.348
	DD vs. (Dd, dd)	AT + AA	98 (84.5%)	99 (79.8%)				
	Overdominant	AT	50 (43.1%)	62 (50%)	1.32 (0.79–2.2)	0.303	1.15	0.283
		TT + AA	66 (56.9%)	62 (50%)				
	Homozygote	TT	18 (27.3%)	25 (40.3%)	1.802 (0.86–3.79)	0.137	2.44	0.118
		AA	48 (72.7%)	37 (59.7%)				
	Heterozygote	AT	50 (73.5%)	62 (71.3%)	0.89 (0.44–1.82)	0.857	0.1	0.752
		TT	18 (26.5%)	25 (28.7%)				
	Allele frequency	A	146 (62.9%)	136 (54.8%)	0.72 (0.5–1.03)	0.078	3.24	0.072
MAF= 0.37	T vs. A	T	86 (37.1%)	112 (45.2%)				

MAF—minor allele frequency, P ^Fi^—*p* value in Fisher’s exact test, P ^Chi^—*p* value in Chi-squared test (for df = 1 or df = 2).

**Table 5 biomedicines-12-00321-t005:** Subgroup analysis: genotypes and allele frequencies between EROC and non-EROC subtypes (n = 193).

SNP	Model	Genotype	Non-EROC	EROC	OR (95% CI)	P ^Fi^	χ^2^	P ^Chi^
**rs4073**	Co-dominant	AA	43 (26.1%)	10 (35.7%)			1.71	0.426
**(−251 A>T)**		AT	86 (52.1%)	11 (39.3%)				
		TT	36 (21.8%)	7 (25%)				
	Dominant	TT + AT	122 (73.9%)	18 (64.3%)	0.63 (0.27–1.48)	0.359	1.12	0.289
		AA	43 (26.1%)	10 (35.7%)				
	Recessive	TT	36 (21.8%)	7 (25%)	1.19 (0.47–3.03)	0.8060	0.14	0.708
		AA + AT	129 (78.2%)	21 (75%)				
	Overdominant	AT	86 (52.1%)	11 (39.3%)	0.59 (0.26–1.35)	0.226	1.58	0.208
		AA + TT	79 (47.9%)	17 (60.7%)				
	Homozygote	AA	43 (54.4%)	10 (58.8%)	1.2 (0.41–3.46)	0.793	0.11	0.740
	(TT vs. AA)	TT	36 (45.6%)	7 (41.2%)				
	Heterozygote	AT	86 (70.5%)	11 (61.1%)	0.66 (0.24–1.83)	0.585	0.65	0.420
	(AT vs. TT)	TT	36 (29.5%)	7 (38.9%)				
	Allele frequency	A	172 (52.1%)	31 (55.4%)	0.88 (0.5–1.55)	0.667	0.2	0.655
MAF 0.48	(T vs. A)	T	158 (47.9%)	25 (44.6%)				
**rs2227306**	Co-dominant	CC	43 (26.1%)	8 (28.6%)			2.16	0.339
**(+781 C>T)**		CT	87 (52.7%)	11 (39.3%)				
		TT	35 (21.2%)	9 (32.1%)				
	Dominant	TT + CT	122 (73.9%)	20 (71.4%)	0.88 (0.36–2.15)	0.818	0.08	0.777
		CC	43 (26.1%)	8 (28.6%)				
	Recessive	TT	35 (21.2%)	9 (32.1%)	1.76 (0.73–4.23)	0.225	1.63	0.201
		CT + CC	130 (78.8%)	19 (67.9%)				
	Overdominant	CT	87 (52.7%)	11 (39.3%)	0.58 (0.26–1.31)	0.223	1.73	0.188
		TT + CC	78 (47.3%)	17 (60.7%)				
	Homozygote	TT	35 (44.9%)	9 (52.9%)	1.38 (0.48–3.96)	0.599	0.37	0.543
		CC	43 (55.1%)	8 (47.1%)				
	Heterozygote	CT	87 (71.3%)	11 (55%)	0.49 (0.19–1.29)	0.191	2.14	0.143
	(CT vs. TT)	TT	35 (28.7%)	9 (45%)				
	Allele frequency	C	173 (52.4%)	27 (48.2%)	1.18 (0.67–2.09)	0.566	0.34	0.559
MAF = 0.48	T vs. C	T	157 (47.6%)	29 (51.8%)				
**rs2227543**	Co-dominant	CC	47 (28.5%)	8 (28.6%)			0.85	0.653
**(+1633 C>T)**		CT	83 (50.3%)	12 (42.9%)				
		TT	35 (21.2%)	8 (28.6%)				
	Dominant	TT + CT	118 (71.5%)	20 (71.4%)	1 (0.41–2.42)	1	0	1
		CC	47 (28.5%)	8 (28.6%)				
	Recessive	TT	35 (21.2%)	8 (28.6%)	1.49 (0.60–3.66)	0.461	0.75	0.386
		CT + CC	130 (78.8%)	20 (71.4%)				
	Overdominant	CT	83 (50.3%)	12 (42.9%)	0.74 (0.33–1.66)	0.542	0.53	0.467
		TT + CC	82 (49.7%)	16 (57.1%)				
	Homozygote	TT	35 (42.7%)	8 (50%	1.34 (0.46–3.93)	0.784	0.29	0.590
		CC	47 (57.3%)	8 (50%)				
	Heterozygote	CT	83 (70.3%)	12 (60%)	0.63 (0.24–1.68)	0.434	0.85	0.357
		TT	35 (29.7%)	8 (40%)				
	Allele frequency	C	177 (53.6%)	28 (50%)	0.86 (0.49–1.52)	0.665	0.25	0.617
MAF = 0.46	T vs. C	T	153 (46.4%)	28 (50%)				
**rs1126647**	Co-dominant	AA	50 (30.3%)	5 (17.9%)			6.24	**0.044**
**(+2767 A>T)**		AT	84 (50.9%)	12 (42.9%)				
		TT	31 (18.8%)	11 (39.3%)				
	Dominant	TT + AT	115 (69.7%)	23 (82.1%)	2 (0.72–5.56)	0.257	1.82	0.177
		AA	50 (30.3%)	5 (17.9%)				
	Recessive	TT	31 (18.8%)	11 (39.3%)	2.8 (1.19–6.56)	0.024	5.91	**0.015**
		AT + AA	134 (81.2%)	17 (60.7%)				
	Overdominant	AT	84 (50.9%)	12 (42.9%)	0.72 (0.32–1.62)	0.541	0.62	0.431
		TT + AA	81 (49.1%)	16 (57.1%)				
	Homozygote	TT	31 (38.3%)	11 (68.8%)	3.55 (1.13–11.18)	0.03	5.06	**0.024**
		AA	50 (61.7%)	5 (31.2%)				
	Heterozygote	AT	84 (73%)	12 (52.3%)	0.40 (0.16–1.01)	0.0797	3.94	**0.047**
		TT	31 (27%)	11 (47.8%)				
	Allele frequency	A	184 (55.8%)	22 (39.3%)	1.95 (1.09–3.47)	0.029	5.22	**0.022**
MAF= 0.44	T vs. A	T	146 (44.2%)	34 (60.7%)				

MAF—minor allele frequency, P ^Fi^**—***p* value in Fisher’s exact test, P ^Chi^**—***p* value in Chi-squared test (for df = 1 or df = 2). Please note that the definitions of the dominant and recessive models of rs4073 changed, as the MAF changed according to the changed distribution of the A and T alleles in the subgroup analysis.

## Data Availability

The data presented in this study are available on reasonable request from the first (R.W.) or the corresponding (E.O.) author.

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
