# Peer review of "Association of Four Interleukin-8 Polymorphisms (−251 A>T, +781 C>T, +1633 C>T, +2767 A>T) with Ovarian Cancer Risk: Focus on Menopausal Status and Endometriosis-Related Subtypes"

_biomedicines, 2024, doi:10.3390/biomedicines12020321_

Round 1

Reviewer 1 Report

Comments and Suggestions for Authors

The manuscript “Association of four interleukin-8 polymorphisms (-251 A>T, +781 C>T, +1633 C>T, +2767 A>T) with ovarian cancer risk: focus on menopausal status and endometriosis-related subtypes” by RafaÅ‚ Watrowski and co-authors to investigate four common IL-8 SNPs: rs4073 (-251 A>T), rs2227306 (+781 C>T), rs2227543 (+1633 C>T), and rs1126647 (+2767 A>T) using the restriction fragment length polymorphism (PCR-RFLP) technique. This study included a cohort of 413 women of Central European descent, consisting of 200 OC patients and 213 healthy controls. The most common (73.5%) histological type was high-grade serous OC (HGSOC), whereas 28/200 (14%) patients had endometriosis-related (clear cell or endometrioid) OC subtypes (EROC). In post-menopausal women, three of four investigated SNPs: rs4073 (-251 A>T), rs2227306 (+781 C>T), and rs2227543 (+1633 C>T) were associated with OC risk. Furthermore, we are the first to report a significant relationship between the T allele or TT genotype of SNP rs1126647 (+2767 A>T) and the EROC subtype (p=0.02 in the co-dominant model). The TT homozygotes were found more than twice in EROC compared to other OC subtypes (39% vs. 19%, p=0.015). None of the examined SNPs appeared to influence OC risk in premenopausal women, nor were they associated with the aggressive HGSOC subtype or the stage of disease at the initial diagnosis. However, some concerns that must be taken into account before the work can be reconsidered for publication.

Comment

1.      Table 1: The gels of IL-8 SNPs PCR-RFLP should be provided in supplementary.

2.      Table 2: Please provide the detailed demographic information of patients, such as: histologic type, stage, with/without chemotherapy.

3.      Please discuss the correlation of rs2227543 (+1633 C>T) and rs1126647 (+2767 A>T) with IL-8 function?

Comments on the Quality of English Language

Moderate editing of English language required.

Author Response

Dear Reviewer,

thank you very much for your comments and suggestions. Following in detail your suggestions, we improved the manuscript by addressing your questions and adding four new figures and one new table.

Reviewer’s Comments:

  1. Table 1: The gels of IL-8 SNPs PCR-RFLP should be provided in supplementary.

Response: We added four figures (Figure A2-A5) presenting electropherograms for each SNP, with each figure containing three sub-pictures (e.g., 2a, 2b, and 2c) representative for each genotype.

  1. Table 2: Please provide the detailed demographic information of patients, such as: histologic type, stage, with/without chemotherapy.

Response: The required data were already partially available in the Appendix. Now, we expanded the information by adding detailed info on FIGO stages in Table A2. The requested information is provided in Table A1, Table A2, and Figure A1.

  1. Please discuss the correlation of rs2227543 (+1633 C>T) and rs1126647 (+2767 A>T) with IL-8 function?

Response: We adressed this issue in the newly add paragraph of the Introduction (line 134-146).

With kind regards,

Rafał Watrowski

Reviewer 2 Report

Comments and Suggestions for Authors

The manuscript is well written, I recommend acceptance in the presence form. It is related to Interleukin-8 (IL-8) that plays a crucial role in regulating inflammation and carcinogenesis. Single Nucleotide Polymorphisms within the IL-8 gene impacting the susceptibility to various cancers such as lung, gastric, and hepatocellular carcinoma. This investigation delves into four common IL-8 SNPs among 413 Central European women, comprising 200 OC patients and 213 healthy controls. A  novel finding reveals a significant correlation between the T allele or TT genotype of SNP rs1126647 and the endometriosis-related OC subtype (EROC). TT homozygotes were notably higher in EROC compared to other subtypes, emphasizing the potential relevance of IL-8 genetic variations in OC pathogenesis. These SNPs showed no discernible impact on OC risk in premenopausal women, nor did they exhibit associations with the aggressive high-grade serous OC subtype or the disease stage at initial diagnosis.

Author Response

Dear Reviewer,

Thank you very much for your favorable review. Your recommendation to accept the manuscript as it is reflects the highest level of appreciation for our work. We are encouraged by your positive feedback and will strive to maintain or even surpass this level of quality in our future research endeavors.

Best regards,

The Authors